# DG-SLAM: Robust Dynamic Gaussian Splatting SLAM with Hybrid Pose Optimization

**Yueming Xu**[1*]    **Haochen Jiang**[1*]    **Zhongyang Xiao**[2]    **Jianfeng Feng**[1]    **Li Zhang**[1†]

[1]Fudan University    [2]Autonomous Driving Division, NIO

https://github.com/fudan-zvg/DG-SLAM

## Abstract

Achieving robust and precise pose estimation in dynamic scenes is a significant research challenge in Visual Simultaneous Localization and Mapping (SLAM). Recent advancements integrating Gaussian Splatting into SLAM systems have proven effective in creating high-quality renderings using explicit 3D Gaussian models, significantly improving environmental reconstruction fidelity. However, these approaches depend on a static environment assumption and face challenges in dynamic environments due to inconsistent observations of geometry and photometry. To address this problem, we propose DG-SLAM, the first robust dynamic visual SLAM system grounded in 3D Gaussians, which provides precise camera pose estimation alongside high-fidelity reconstructions. Specifically, we propose effective strategies, including motion mask generation, adaptive Gaussian point management, and a hybrid camera tracking algorithm to improve the accuracy and robustness of pose estimation. Extensive experiments demonstrate that DG-SLAM delivers state-of-the-art performance in camera pose estimation, map reconstruction, and novel-view synthesis in dynamic scenes, outperforming existing methods meanwhile preserving real-time rendering ability.

## 1 Introduction

Visual simultaneous localization and mapping (SLAM), the task of reconstructing a 3D map within an unknown environment while simultaneously estimating the camera pose, is recognized as a critical component to achieving autonomous navigation in novel 3D environments for mobile robots [1]. It has been widely used in various forms in fields such as robotics, autonomous driving, and augmented/virtual reality (AR/VR). However, the majority of previous research [2, 3, 4, 5, 6, 7, 8, 9, 10] typically relies on the assumption of static environments, limiting the practical applicability of this technology in daily life. Consequently, how to achieve accurate and robust pose estimation in dynamic scenes remains an urgent problem to be addressed for mobile robots.

Recently, many researchers [6, 7, 8, 9, 10] have endeavored to replace the conventional explicit representations used in visual SLAM, such as Signed Distance Function (SDF), voxel grids [11], meshes [12], and surfel clouds [13], with the neural radiance field (NeRF) [14] approach for reconstructing the neural implicit map. This novel map representation is more continuous, efficient, and able to be optimized with differentiable rendering, which has the potential to benefit applications like navigation and reconstruction. However, these methods exhibit two primary issues: the scene's bounds are required to be predefined to initialize the neural voxel grid; and the implicit representation proves challenging for information fusion and editing. To address these problems, recent works

---

*Yueming Xu and Haochen Jiang contribute equally to this work.

†Li Zhang (lizhangfd@fudan.edu.cn) is the corresponding author with School of Data Science, Fudan University.

38th Conference on Neural Information Processing Systems (NeurIPS 2024).

like GS-SLAM [15], SplaTam [16], and Gaussian splatting SLAM [17] leverage the 3D-GS [18] to explicit represent the scene's map. This explicit geometric representation is also smooth, continuous, and differentiable. Moreover, a substantial array of Gaussians can be rendered with high efficiency through splatting rasterization techniques, achieving up to 300 frames per second (FPS) at a resolution of 1080p. However, all these above-mentioned neural SLAM methods do not perform well in dynamic scenes. The robustness of these systems significantly decreases, even leading to tracking failures, when dynamic objects appear in the environment.

To tackle these problems, we propose a novel 3D Gaussian-based visual SLAM that can reliably track camera motion in dynamic indoor environments. Due to the capability of 3D-GS to accomplish high-quality rendering in real-time, the SLAM system more readily converges to a global optimum during pose optimization, thereby achieving better and more stable pose optimization results. A cornerstone of our approach to achieving robust pose estimation lies in the innovative motion mask generation algorithm. This algorithm filters out sampled pixels situated within invalid zones, thereby refining the estimation process. In addition to the constraint of depth residual, we employ a spatio-temporal consistency strategy within an observation window to generate depth warp masks. By incrementally fusing the depth warp mask and semantic mask, the motion mask will become more precise to reflect the true motion state of objects. To improve the accuracy and stability of pose estimation, we leverage DROID-SLAM [19] odometry (DROID-VO) to provide an initial pose estimate and devise a coarse-to-fine optimization algorithm built upon the initially estimated camera pose. This aims to minimize the disparity between pose estimation and the reconstructed map, employing photorealistic alignment optimization through Gaussian Splatting. Moreover, this hybrid pose optimization approach effectively ensures the accuracy and quality of the generated depth warp mask, thereby facilitating better performance in the next camera tracking stage. To obtain high-quality rendering results, we propose a novel adaptive Gaussian point addition and pruning method to keep the geometry clean and enable accurate and robust camera tracking. Capitalizing on the factor graph structure inherent in DROID-SLAM, our system is capable of executing dense Bundle Adjustment (DBA) upon completion of tracking to eliminate accumulated errors.

In summary, our **contributions** are summarized as follows: **(i)** To the best of our knowledge, this is the first robust dynamic Gaussian splatting SLAM with hybrid pose optimization, capable of achieving both real-time rendering and high-fidelity reconstruction performance. **(ii)** To mitigate the impact of dynamic objects during pose estimation, we propose an advanced motion mask generation strategy that integrates spatio-temporal consistent depth masks with semantic priors, thereby significantly enhancing the precision of motion object segmentation. **(iii)** We design a hybrid camera tracking strategy utilizing the coarse-to-fine pose optimization algorithm to improve the consistency and accuracy between the estimated pose and the reconstructed map. **(iv)** To better manage and expand the Gaussian map, we propose an adaptive Gaussian point addition and pruning strategy. It ensures geometric integrity and facilitates accurate camera tracking. **(v)** Extensive evaluations on two challenging dynamic datasets and one common static dataset demonstrate the state-of-the-art performance of our proposed SLAM system, particularly in real-world scenarios.

## 2 Related work

**Visual SLAM with dynamic objects filter.** Dynamic object filtering is pivotal for reconstructing static scenes and bolstering the robustness of pose estimation. Existing approaches fall into two main categories: the first relies on re-sampling and residual optimization strategies to remove outliers, as seen in works such as ORB-SLAM2 [2], ORB-SLAM3 [3], and Refusion [5]. These methods, however, are generally limited to addressing small-scale motions and often falter in the face of extensive, continuous object movements. The second group employs the additional prior knowledge, for example, semantic segmentation or object detection prior [20, 21, 22, 23, 24, 25, 26], to remove the dynamic objects. However, all these methods often exhibit a domain gap when applied in real-world environments, leading to the introduction of prediction errors. More recently, deep neural networks have been employed to build end-to-end visual odometry, which performs better in specific environments such as DROID-SLAM [19], DytanVO [27], and DeFlowSLAM [28]. However, these methods still require a substantial amount of training data and are unable to reconstruct the high-fidelity static map.

**RGB-D SLAM with neural implicit representation.** Neural implicit scene representations, also known as neural fields [29], have attracted considerable attention in the field of RGB-D SLAM for

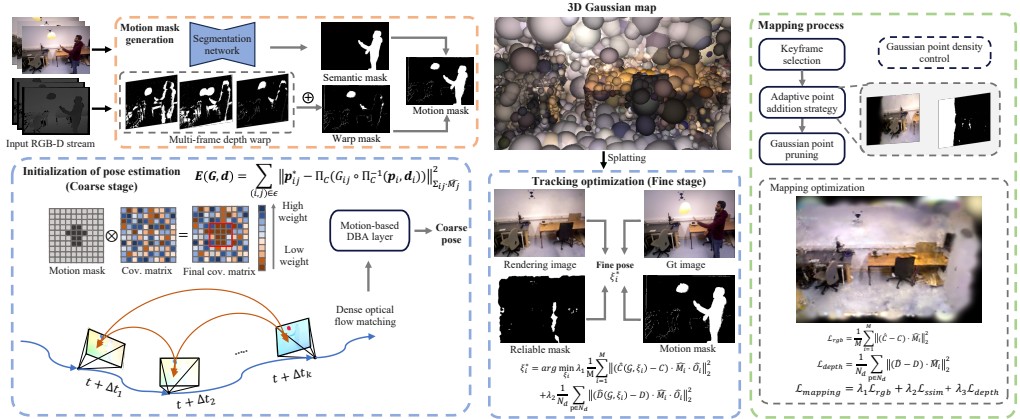

Figure 1: **Overview of DG-SLAM.** Given a series of RGB-D frames, we reconstruct the static high-fidelity 3D Gaussian map and optimize the camera pose represented with lie algebra $\xi_i$.

their impressive expressiveness and low memory footprint. Initial studies, including iMap [6] and DI-Fusion [30], explored the utilization of a single MLP and a feature grid to encode scene geometries within a latent space. However, they both share a critical issue: the problem of network forgetting, which is catastrophic for long-term localization and mapping. In response to this limitation, NICE-SLAM [7] introduces a hierarchical feature grid approach to enhance scene representation fidelity and implements a localized feature updating strategy to address the issue of network forgetting. While these advancements contribute to improved accuracy, they necessitate greater memory consumption and may impact the system's ability to operate in real-time. More recently, existing methods like Vox-Fusion [10], Co-SLAM [8], and ESLAM [9] explore sparse encoding or tri-plane representation strategy to improve the quality of scene reconstruction and the system's execution efficiency. Point-SLAM [31] draws inspiration from the concept of Point-NeRF [32], utilizing neural points to encode spatial geometry and color features. It employs an explicit method to represent spatial maps, effectively improving the accuracy of localization and mapping. All these methods have demonstrated impressive results based on the strong assumptions of static scene conditions. The robustness of these systems significantly decreases when dynamic objects appear in the environment. Recently, Rodyn-SLAM [33] proposed utilizing optical flow and semantic segmentation prior to filter out dynamic objects, and employing a neural rendering method as the frontend. However, this approach is computationally intensive and limits the maximum accuracy achievable in pose estimation.

**3D Gaussian splatting SLAM methods.** Compared to the aforementioned NeRF-based SLAM methods, 3D Gaussian splatting (3D-GS) [18] has garnered widespread interest among researchers due to its advantages in high-fidelity and real-time rendering. Unlike previous implicit map representation methods, 3D-GS explicitly models scene maps by independent Gaussian spheres, naturally endowing it with geometric structure property. Some researchers [17, 15, 34, 16, 35] are exploring the replacement of implicit representations (NeRF) with 3D-GS in the mapping thread. However, these methods are currently constrained by the assumption of static environments, rendering them ineffective in dynamic scenes. It significantly restricts the practical application of Gaussian SLAM systems in real-life scenarios. Under the premise of ensuring high-fidelity reconstruction and real-time rendering, our system is designed to improve the accuracy and robustness of pose estimation under dynamic environments.

## 3 Approach

Given a sequence of RGB-D frames $\{I_i, D_i\}_{i=1}^N, I_t \in \mathbb{R}^3, D_t \in \mathbb{R}$, our method (Fig. 1) aims to simultaneously recover camera poses $\{\xi_i\}_{i=1}^N, \xi_t \in \mathbb{SE}(3)$ and reconstruct the static 3D scene map represented by 3D Gaussian sphere in dynamic environments. Similar to most modern SLAM systems [36, 37], our system comprises two distinct processes: the tracking process as the frontend and the mapping process as the backend.

### 3.1 3D Gaussian map representation

To obtain real-time rendering and high-fidelity reconstruction mapping, we represent the scene as a set of 3d Gaussian ellipsoid $\mathcal{G}$, which simultaneously possesses geometric and appearance properties.

$$\mathcal{G} = \{\mathcal{G}_i : (\mu_i, \boldsymbol{\Sigma}_i, \alpha_i, \mathbf{h}_i) \big| \forall \mathcal{G}_i \in \mathcal{G}\} \tag{1}$$

Each 3D Gaussian ellipsoid $\mathcal{G}_i$ is composed of its center position $\mu_i \in \mathbb{R}^3$, covariance matrix $\boldsymbol{\Sigma}_i \in \mathbb{R}^{3\times3}$, opacity $\alpha_i \in \mathbb{R}$, and spherical harmonics coefficients $\mathbf{h}_i \in \mathbb{R}^{16}$.

**Color and depth splatting rendering.** To obtain the rendering image of color and depth, we project the 3D Gaussian $(\mu_w, \boldsymbol{\Sigma}_w)$ in world coordinate to 2D Gaussian $(\mu_I, \boldsymbol{\Sigma}_I)$ on the image plane:

$$\mu_I = \pi\left(\boldsymbol{T}_w^c \cdot \mu_w\right), \quad \boldsymbol{\Sigma}_I = \mathbf{J}\mathbf{R}\boldsymbol{\Sigma}_w\mathbf{R}^T\mathbf{J}^T, \tag{2}$$

where $\mathbf{R}$ is the viewing transformation and $\mathbf{J}$ is the Jacobian of the affine approximation of the projective transformation. Following the alpha blending method in 3DGS [18], we accumulate the splatting Gaussian ellipsoid along the observation image pixel at the current estimation pose $\xi_i$ to render the color and depth as:

$$\hat{C} = \sum_{i\in\mathcal{G}} \mathbf{c}_i f_i(\mu_I, \boldsymbol{\Sigma}_I) \prod_{j=1}^{i-1}(1 - f_j(\mu_I, \boldsymbol{\Sigma}_I)), \quad \hat{D} = \sum_{i\in\mathcal{G}} \mathbf{d}_i f_i(\mu_I, \boldsymbol{\Sigma}_I) \prod_{j=1}^{i-1}(1 - f_j(\mu_I, \boldsymbol{\Sigma}_I)), \tag{3}$$

where $f_i(\cdot)$ is the Gaussian distribution function weighted by opacity $\alpha_i$. $\mathbf{c}_i$ represents the color of the projected Gaussian point computed by learnable spherical harmonics coefficients $\mathbf{h}_i$. Similarly, $\mathbf{d}_i$ denotes the depth of Gaussian point $\mathcal{G}_i$, obtained by projecting to z-axis in the camera coordinate.

**Accumulated opacity.** We use accumulated opacity $\hat{O}$ to represent the rendering reliability for each pixel and judge whether the Gaussian map is well-optimized.

$$\hat{O} = \sum_{i\in\mathcal{G}} f_i(\mu_I, \boldsymbol{\Sigma}_I) \prod_{j=1}^{i-1}(1 - f_j(\mu_I, \boldsymbol{\Sigma}_I)), \tag{4}$$

### 3.2 Motion mask generation

For each input keyframe, we select its associated keyframe set $\mathcal{D}$ within a sliding window. To reduce the computation and improve the accuracy of generating motion mask, we employ the depth warping operation solely on keyframes. To ensure the overlap between adjacent keyframes is not too small, we employ optical-flow distance to determine keyframe insertion. In regions with more intense motion, our goal is to insert as many keyframes as possible.

For the pixel $p$ in keyframe $i$, we reproject it onto keyframe $j$ as follows:

$$\boldsymbol{p}_{i\to j} = f_{warp}\left(\xi_{ji}, \boldsymbol{p}_i, D_i(\boldsymbol{p}_i)\right) = \boldsymbol{K}\boldsymbol{T}_{ji}\left(\boldsymbol{K}^{-1}D_i(\boldsymbol{p}_i)\boldsymbol{p}_i^{homo}\right), \tag{5}$$

where $\boldsymbol{K}$ and $\boldsymbol{T}_{ji}$ represent the intrinsic matrix and the transformation matrix between frame $i$ and frame $j$, respectively. $\boldsymbol{p}_i^{homo} = (u, v, 1)$ is the homogeneous coordinate of $\boldsymbol{p}_i$.

Given any associate keyframes $D_i, D_j \in \mathcal{D}$, we utilize warp function $f_{warp}$ to compute the residual of reprojection depth value. By setting a suitable threshold $e_{th}$, we derive the depth warp mask $\widehat{\mathcal{M}}_{j,i}^{wd}$ corresponding to dynamic objects as:

$$\widehat{\mathcal{M}}_{j,i}^{wd} : \left\{\bigcap_{\boldsymbol{p}_i\in D_i} \mathbf{1}(D_j(\boldsymbol{p}_{i\to j}) - D_i(\boldsymbol{p}_i) < e_{th}) \otimes \boldsymbol{I}_{m\times n}\right\} \tag{6}$$

where $\boldsymbol{I}_{m\times n}$ represents a matrix of the same size as the image, filled with ones. The $\otimes$ operation signifies that for each element in the matrix $\boldsymbol{I}_{m\times n}$, we assess whether its warp depth meets a specified threshold and subsequently modify the corresponding value at that position. As illustrated in Fig. 1, to derive a more precise warp mask, we consider the spatial coherence of object motion within a sliding window of length $N$ and combine the multiple observation warp masks. Unlike ReFusion [5], our method can mitigate the potential impact of observation noise from a single warp mask. When object motion becomes significant, we only mask the foreground pixels where the depth residual is positive to avoid a large portion of pixels being masked as dynamic regions. Subsequently, we integrate the warp mask and semantic mask to derive the final motion mask $\widehat{\mathcal{M}}_j$ as:

$$\widehat{\mathcal{M}}_j = \widehat{\mathcal{M}}_{j,i}^{wd} \cap \widehat{\mathcal{M}}_{j,i-1}^{wd} \cap \widehat{\mathcal{M}}_{j,i-2}^{wd} \cdots \cap \widehat{\mathcal{M}}_{j,i-N}^{wd} \cup \widehat{\mathcal{M}}_j^{sg}, \tag{7}$$

Thanks to our innovative approach to generating motion masks, the omission of dynamic objects by semantic priors can be effectively compensated. Furthermore, by leveraging a spatial consistency strategy, the inaccuracy of edge region recognition during depth warping can be significantly reduced.

### 3.3 Coarse-to-fine camera tracking.

The constant speed motion model struggles to infer a reasonable initial optimized pose value in dynamic scenes. An inaccurate initial pose will lead to optimization getting trapped in local optima more easily and can affect the generation quality of the depth warp mask. To achieve more precise pose estimation in dynamic environments, we utilize the visual odometry (VO) component from DROID-SLAM [19] as the coarse pose estimation results in camera tracking. We also conduct a dense bundle adjustment in every interaction for a set of keyframes to optimize the corresponding pose $\mathbf{G}$ and depth $\mathbf{d}$:

$$\mathbf{E}(\mathbf{G}, \mathbf{d}) = \sum_{(i,j) \in \epsilon} \left\| \mathbf{p}_{ij}^* - \Pi_C \left( G_{ij} \circ \Pi_C^{-1} \left( \mathbf{p}_i, \mathbf{d}_i \right) \right) \right\|_{\Sigma_{ij} \cdot \widehat{\mathcal{M}}_j}^2, \tag{8}$$

where $\Sigma_{ij} = \text{diag}(w_{ij})$, $w_{ij}$ represents the confidence weights. $G_{ij}$ denotes the relative transformation between the poses $G_i$ and $G_j$. $\mathbf{p}_i$ means a grid of pixel coordinates. Moreover, $\mathbf{p}_{ij}^*$ is corrected correspondence as predicted by updated optical flow estimation. To overcome the influence of dynamic objects on bundle adjustment, we introduce suppression through the motion mask $\widehat{\mathcal{M}}_j$ applied to the weighted covariance matrix. Consequently, the weighted confidence associated with dynamic objects is reduced to zero.

To further improve the accuracy of pose estimation and minimize inconsistencies between the estimated pose and the reconstructed map, we implement fine camera tracking by leveraging Gaussian splatting, based on the initial obtained pose. Due to obstructions by dynamic objects and the inadequate optimization of Gaussian points in the map, the rendered image may exhibit blurring or black floaters. Therefore, we employ the accumulated opacity, as outlined in 4, to denote the pixel rendering reliability. The reliable mask $\widehat{\mathcal{O}}_i$ for camera tracking is generated as follows:

$$\widehat{\mathcal{O}}_i : \left\{ \bigcap_{[\mu_j, v_j] \in I_i} \mathbf{1}(\hat{O}[\mu_j, v_j] > \tau_{track}) \otimes \boldsymbol{I}_{m \times n} \right\}, \tag{9}$$

where $\mu_j, v_j$ represents the pixel location. When the accumulated opacity at the pixel exceeds a given threshold $\tau_{track}$, we consider the Gaussian point associated with that pixel to be well-optimized. Consequently, using the rendered image at these pixels for pose estimation is deemed reliable. The overall loss function is finally formulated as the following minimization:

$$\xi_i^* = \arg\min_{\xi_i} \lambda_1 \frac{1}{M} \sum_{i=1}^M \left\| \left( \hat{C}(\mathcal{G}, \xi_i) - C \right) \cdot \widehat{\mathcal{M}}_i \cdot \widehat{\mathcal{O}}_i \right\|_2^2 + \lambda_2 \frac{1}{N_d} \sum_{\mathbf{p} \in N_d} \left\| \left( \hat{D}(\mathcal{G}, \xi_i) - D \right) \cdot \widehat{\mathcal{M}}_i \cdot \widehat{\mathcal{O}}_i \right\|_2^2. \tag{10}$$

Where $\xi_i$ denotes the camera pose requiring optimization. $C$ and $D$ denote the ground truth color and depth map, respectively. $M$ represents the number of sampled pixels in the current image. Note that only pixels with valid depth value $N_d$ are considered in optimization.

This hybrid pose optimization approach effectively ensures the accuracy and quality of the warp mask, thereby facilitating better performance in the next camera tracking stage. In short, this hybrid pose optimization strategy enables us to achieve more precise and robust pose estimation whether in dynamic or static environments.

### 3.4 SLAM system

**Map initialization.** For the first frame, we do not conduct the tracking step and the camera pose is set to the identity matrix. To better initialization, we reduce the gradient-based dynamic radius to half so that can add more Gaussian points. For pixels located outside the motion mask, we sample and reproject them to the world coordinates. We initialize the Gaussian point color and center position $\mu_i$ with the RGB value and reprojection coordinate of the pixel, respectively. The opacity $\alpha_i$ is set as $0.1$ and the scale vector $\mathbf{S}_i$ is initialized based on the mean distance of the three nearest neighbor points. The rotation matrix $\mathbf{R}_i$ is initialized as the identity matrix.

**Map optimization.** To optimize the scene Gaussian representation, we render depth and color in independent view as seen Eq. 3, comparing with the ground truth map:

$$\mathcal{L}_{rgb} = \frac{1}{M} \sum_{i=1}^M \left\| \left( \hat{C} - C \right) \cdot \widehat{\mathcal{M}}_i \right\|_2^2, \quad \mathcal{L}_{depth} = \frac{1}{N_d} \sum_{\mathbf{p} \in N_d} \left\| \left( \hat{D} - D \right) \cdot \widehat{\mathcal{M}}_i \right\|_2^2, \tag{11}$$

In contrast to existing methods, we introduce the motion mask $\widehat{\mathcal{M}}_i$ to remove sampled pixels within the dynamic region effectively. The final Gaussian map optimization is performed by minimizing the

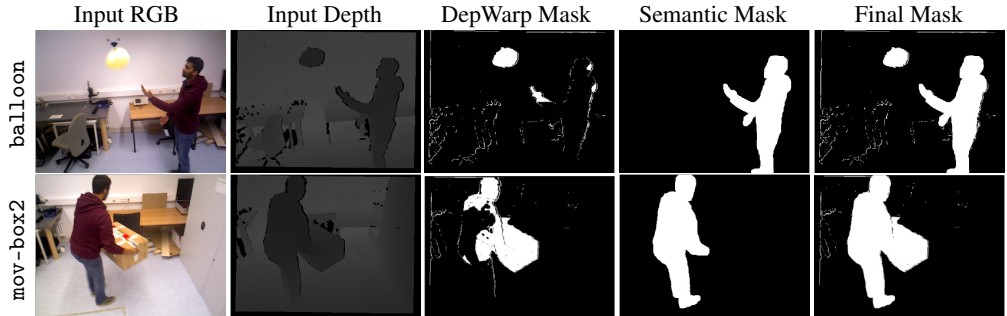

Figure 2: **Qualitative results of the motion mask generation.** By fusing the semantic mask and depth warp mask, the final mask will be more precise.

geometric depth loss and photometric color loss :

$$\mathcal{L}_{mapping} = \lambda_1 \mathcal{L}_{rgb} + \lambda_2 \mathcal{L}_{ssim} + \lambda_3 \mathcal{L}_{depth}, \tag{12}$$

where $\mathcal{L}_{ssim}$ denotes the structural similarity loss between two images. Moreover, $\lambda_1, \lambda_2, \lambda_3$ are weight factors for balance in the optimization process.

**Adapative Gaussian point adding strategy.** To guarantee the superior scene representation capability of 3D Gaussian, we also employ dynamic Gaussian point density, which is inspired by Point-SLAM [31]. The insertion radius is determined based on color gradient, allowing for a denser allocation of points in areas with high texture while minimizing point addition in regions of low texture. This method efficiently manages memory consumption by adapting the point density according to the textural complexity of the scene.

To ensure the points we add are both necessary and effective, we adopt a two-stage adaptive point-add strategy. For new viewpoints without previous observation, we initially perform uniform sampling across the entire image to ensure that new observed areas can be covered by Gaussian points. Moreover, if the accumulated opacity $\hat{O}$, calculated by Eq. 4, falls below the threshold $o_{th}$, or the depth residual between the rendered pixels and the ground truth depth is excessively large, we then add 3D Gaussian points based on these under-fitting pixels. At last, these new Gaussian points will be initialized based on the point density.

**Map point deleting strategy.** Given that the added 3D Gaussian points have not been subjected to geometric consistency verification and may exhibit unreasonable representation values during optimization, this could lead to the generation of a low-quality dense map or the introduction of numerous artifacts in the rendering image. we implement the pruning operation as part of the Gaussian map optimization. To ensure the points we delete are both reasonable and accurate, we also adopt a two-stage point delete strategy. For all Gaussian points observed from the current viewpoint, we delete points based on three criteria: the opacity value, the maximum scale, and the ratio between the ellipsoid's major and minor axes, described as follow:

$$\alpha_i < \tau_\alpha \quad \text{or} \quad \max(\mathbf{S}) > \tau_{s1} \quad \text{or} \quad \frac{\max(\mathbf{S})}{\min(\mathbf{S})} > \tau_{s2}. \tag{13}$$

Moreover, due to potential inaccuracies along the edges of the current motion mask, adding these points could result in artifacts within the scene map. Thus, we project these points on keyframes in a small sliding window to recheck whether they can be observed by these keyframes. If the number of observations of a point is too low, we consider its addition to be insufficiently robust, and therefore, it will be removed from the current Gaussian map.

## 4 Experiments

**Datasets.** Our methodology is evaluated using three publicly available challenging datasets: *TUM RGB-D* dataset [38], *BONN RGB-D Dynamic* dataset [5] and *ScanNet* [39]. These three datasets contain both challenging dynamic scenes and real static scenes. This selection facilitates a comprehensive assessment of our approach under varied conditions, demonstrating its applicability and robustness in real-world indoor environments.

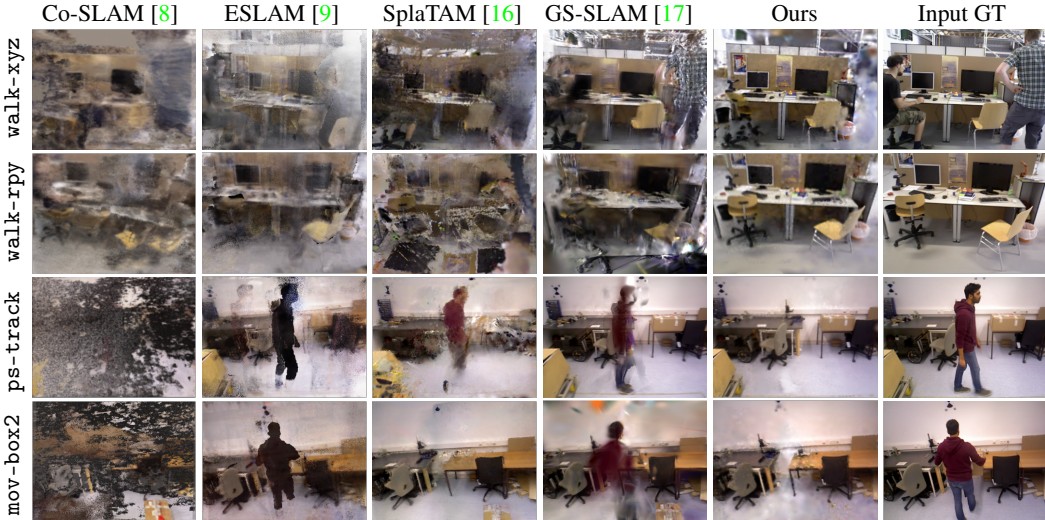

|  | Co-SLAM [8] | ESLAM [9] | SplaTAM [16] | GS-SLAM [17] | Ours | Input GT |

Figure 3: **Visual comparison of the rendering image on the *TUM* and *BONN* datasets.** Our results are more complete and accurate without the dynamic object floaters.

|  |  | ball | ball2 | ps_trk | ps_trk2 | mv_box2 | Avg. |
|---|---|---|---|---|---|---|---|
| NICE-SLAM[7] | **Acc.**[cm]↓ | X | 24.30 | 43.11 | 74.92 | 17.56 | 39.97 |
|  | **Comp.**[cm]↓ | X | 16.65 | 117.95 | 172.20 | 18.19 | 81.25 |
|  | **Comp. Ratio**[≤ 5cm%]↑ | X | 29.68 | 15.89 | 13.96 | 32.18 | 22.93 |
| Co-SLAM[8] | **Acc.**[cm]↓ | 10.61 | 14.49 | 26.46 | 26.00 | 12.73 | 18.06 |
|  | **Comp.**[cm]↓ | 10.65 | 40.23 | 124.86 | 118.35 | 10.22 | 60.86 |
|  | **Comp. Ratio**[≤ 5cm%]↑ | 34.10 | 3.21 | 2.05 | 2.90 | 39.10 | 16.27 |
| ESLAM[9] | **Acc.**[cm]↓ | 17.17 | 26.82 | 59.18 | 89.22 | 12.32 | 40.94 |
|  | **Comp.**[cm]↓ | **9.11** | 13.58 | 145.78 | 186.65 | 10.03 | 73.03 |
|  | **Comp. Ratio**[≤ 5cm%]↑ | 47.44 | 47.94 | 20.53 | 17.33 | 41.41 | 34.93 |
| DG-SLAM(Ours) | **Acc.**[cm]↓ | **7.00** | **5.80** | **9.14** | **11.78** | **6.56** | **8.06** |
|  | **Comp.**[cm]↓ | 9.80 | **8.05** | **17.99** | **20.10** | **7.61** | **15.46** |
|  | **Comp. Ratio**[≤ 5cm%]↑ | **49.46** | **52.41** | **34.62** | **32.81** | **49.02** | **43.67** |

Table 1: **Reconstruction results on several dynamic scene sequences in the *BONN* dataset**. Instances of tracking failures are denoted by "X". The most superior outcomes within the domain of RGB-D SLAMs are highlighted in **bold** for emphasis.

**Metrics.** For the evaluation of pose estimation, we utilize the Root Mean Square Error (RMSE) and Standard Deviation (STD) of Absolute Trajectory Error (ATE) [38]. Prior to assessment, the estimated trajectory is aligned with the ground truth trajectory via Horn's Procrustes method. [40], ensuring a coherent basis for evaluation. To evaluate the reconstruction quality of static maps in dynamic scenes, we employ three metrics(i) *Accuracy* (cm), (ii) *Completion* (cm), and (iii) *Completion Ratio* (percentage of points within a 5cm threshold), following NICE-SLAM [7]. Since the BONN dataset provides only the ground truth point cloud, we randomly sampled 200,000 points from the GT point cloud and the reconstructed mesh surface to calculate these metrics.

**Implementation details.** We run our DG-SLAM on an RTX 3090 Ti GPU at 2 FPS on BONN datasets, which takes roughly 9GB of memory. We set the loss weight $\lambda_1 = 0.9$ , $\lambda_2 = 0.2$ and $\lambda_3 = 0.1$ to train our model. The number of iterations for the tracking and mapping processes has been set to 20 and 40, respectively. For the Gaussian points deleting, we set $\tau_\alpha = 0.005$, $\tau_{S1} = 0.4$ and $\tau_{S2} = 36$ to avoid the generation of abnormal Gaussian points. What's more, we utilize Oneformer [41] to generate prior semantic segmentation. For the depth wrap mask, we set the window size to 4 and the depth threshold to 0.6. We also adopt the keyframe selection strategy from DROID-VO [19] based on optical flow.

| Method | f3/w_r | f3/w_x | f3/w_s | f3/s_x | f2/d_p | f3/l_o | Avg. |
|---|---|---|---|---|---|---|---|
| ORB-SLAM3 [3] | 68.7 | 28.1 | 2.0 | **1.0** | **1.5** | **1.0** | 17.1 |
| ReFusion [5] | - | 9.9 | 1.7 | 4.0 | - | - | 5.2 |
| Co-fusion [42] | - | 69.6 | 55.1 | 2.7 | - | - | 42.5 |
| MID-fusion [43] | - | 6.8 | 2.3 | 6.2 | - | - | 5.1 |
| EM-fusion [44] | - | 6.6 | 1.4 | 3.7 | - | - | 3.9 |
| iMAP*[6] | 139.5 | 111.5 | 137.3 | 23.6 | 119.0 | 5.8 | 89.5 |
| NICE-SLAM[7] | X | 113.8 | 88.2 | 7.9 | X | 6.9 | 54.2 |
| Vox-Fusion[10] | X | 146.6 | 109.9 | 3.8 | X | 26.1 | 71.6 |
| Co-SLAM[8] | 52.1 | 51.8 | 49.5 | 6.0 | 7.6 | 2.4 | 28.3 |
| ESLAM[9] | 90.4 | 45.7 | 93.6 | 7.6 | X | 2.5 | 48.0 |
| Rodyn-SLAM[33] | 7.8 | 8.3 | 1.7 | 5.1 | 5.6 | 2.8 | 5.3 |
| SplaTAM[16] | 100.4 | 218.3 | 115.2 | 1.7 | 5.4 | 5.1 | 74.4 |
| GS-SLAM[17] | 33.5 | 37.7 | 8.4 | 2.7 | 8.6 | 1.8 | 15.5 |
| DROID-VO[19] | 10.0 | 1.7 | 0.7 | 1.1 | 3.7 | 2.3 | 3.3 |
| DG-SLAM(Ours) | **4.3** | **1.6** | **0.6** | **1.0** | 3.2 | 2.3 | **2.2** |

Table 2: **Camera tracking results on several dynamic scene sequences in the *TUM* dataset**. "∗" denotes the version reproduced by NICE-SLAM. "X" and "-" denote the tracking failures and absence of mention, respectively. The metric is Absolute Trajectory Error (ATE) and the unit is [cm].

| Method | ball | ball2 | ps_tk | ps_tk2 | ball_tk | mv_box2 | Avg. |
|---|---|---|---|---|---|---|---|
| ORB-SLAM3 [3] | 5.8 | 17.7 | 70.7 | 77.9 | **3.1** | **3.5** | 29.8 |
| ReFusion [5] | 17.5 | 25.4 | 28.9 | 46.3 | 30.2 | 17.9 | 27.7 |
| iMAP*[6] | 14.9 | 67.0 | 28.3 | 52.8 | 24.8 | 28.3 | 36.1 |
| NICE-SLAM[7] | X | 66.8 | 54.9 | 45.3 | 21.2 | 31.9 | 44.1 |
| Vox-Fusion[10] | 65.7 | 82.1 | 128.6 | 162.2 | 43.9 | 47.5 | 88.4 |
| Co-SLAM[8] | 28.8 | 20.6 | 61.0 | 59.1 | 38.3 | 70.0 | 46.3 |
| ESLAM[9] | 22.6 | 36.2 | 48.0 | 51.4 | 12.4 | 17.7 | 31.4 |
| Rodyn-SLAM[33] | 7.9 | 11.5 | 14.5 | 13.8 | 13.3 | 12.6 | 12.3 |
| SplaTAM[16] | 35.5 | 36.1 | 149.7 | 91.2 | 12.5 | 19.0 | 57.4 |
| GS-SLAM[17] | 37.5 | 26.8 | 46.8 | 50.4 | 31.9 | 4.8 | 33.1 |
| DROID-VO[19] | 5.4 | 4.6 | 21.4 | 46.0 | 8.9 | 5.9 | 15.4 |
| DG-SLAM(Ours) | **3.7** | **4.1** | **4.5** | **6.9** | 10.0 | **3.5** | **5.5** |

Table 3: **Camera tracking results on several dynamic scene sequences in the *BONN* dataset.** "∗" denotes the version reproduced by NICE-SLAM. "X" denotes the tracking failures. The metric is ATE and the unit is [cm].

## 4.1 Evaluation of generating motion mask

We evaluated our method on the `balloon` and `move_no_box2` sequences of the *BONN* dataset to show the qualitative results of the generated motion mask. In these two sequences, in addition to the typical motion of pedestrians, there are movements of atypical objects accompanying the human, such as balloons and boxes. It might be missed if we rely solely on the semantic prior. As shown in Fig. 2, our generated methods notably improve the precision of motion mask segmentation, effectively reducing the inconsistencies along edge regions and detecting the true dynamic objects.

## 4.2 Evaluation of mapping performance

To more effectively showcase the performance of our system in dynamic environments, we evaluate the reconstruction results from both qualitative and quantitative perspectives. Given dynamic scene datasets seldom offer static GT mesh or point cloud, we utilize the *BONN* dataset for our quantitative analytical experiments. We compare our DG-SLAM method against current state-of-the-art neural-based SLAM methods, all of which are available as open-source projects.

As shown in Tab. 1, our method significantly surpasses contemporary approaches in terms of accuracy, completion, and completion ratio metrics, achieving state-of-the-art performance. Meanwhile, the reconstructed static map can be rendered with high fidelity, as shown in Fig. 3. This also indirectly demonstrates that our methods can generate a more accurate static map compared with other mainstream SLAM systems.

## 4.3 Evaluation of tracking performance

To comprehensively assess the tracking performance of our DG-SLAM approach, we conduct comparative analyses within highly dynamic, slightly dynamic, and static environments. The comparison methods encompass classical SLAM methods such as ORB-SLAM3 [3], ReFusion [5],MID-fusion [43], and EM-fusion [44], and widely recognized NeRF-based SLAM systems such as NICE-SLAM [7], iMap [6], Vox-Fusion [10], ESLAM [9], Co-SLAM [8]. We also incorporate a comparison with the newly proposed dynamic neural RGB-D SLAM system, Rodyn-SLAM [33]. Furthermore, our evaluation extends to the latest advancements in 3D Gaussian-based SLAM, including SplaTAM [16] and GS-SLAM [17].

**Dynamic scenes.** As shown in Tab. 2, we report the results on three highly dynamic sequences, two slightly dynamic sequences, and one static sequence from the TUM RGB-D dataset. Our system exhibits exceptional tracking performance, attributed to the implementation of the map point deleting strategy and the powerful coarse-to-fine camera tracking algorithm. Furthermore, the tracking capabilities of our system have also been rigorously evaluated on the intricate and demanding BONN RGB-D dataset, with outcomes presented in Tab. 3. In dynamic scenarios characterized by heightened complexity and challenge, our method has consistently demonstrated superior performance, underscoring its effectiveness and reliability in real-world navigation applications. Meanwhile, we also showcased the number of iterations of the tracking and mapping process, taking TUM as an example, as shown in Tab. 5. Compared to other methods, our method achieved superior results while maintaining competitive iterations and efficiency.

**Static scenes.** To better illustrate the robustness of our system, we also evaluate our methods with existing SLAM on common real-world static sequences from ScanNet [39]. As shown in Tab. 4, our DG-SLAM still achieves competitive performance within static scenes with fewer tracking and mapping iterations, despite our method being designed for dynamic scenes. Notably, the motion mask will become irrelevant in static scenes. Thus, it can sufficiently demonstrate the effectiveness of our proposed hybrid camera tracking strategy and adaptive Gaussian point management strategy.

| Method | 00 | 59 | 106 | 169 | 181 | 207 | Avg. |
|---|---|---|---|---|---|---|---|
| NICE-SLAM [7] | 12.0 | 14.0 | 7.9 | 10.9 | 13.4 | 6.2 | 10.7 |
| Co-SLAM [8] | **7.1** | 11.1 | 9.4 | **5.9** | 11.8 | **7.1** | 8.8 |
| Point-SLAM [31] | 10.2 | 7.8 | 8.7 | 22.2 | 14.8 | 9.5 | 12.2 |
| SplaTAM [16] | 12.8 | 10.1 | 17.7 | 12.1 | 11.1 | 7.5 | 11.9 |
| GS-SLAM [17] | 13.6 | **7.6** | 8.1 | 13.7 | 34.6 | 12.5 | 15.1 |
| DG-SLAM(Ours) | 7.9 | 11.5 | **8.0** | 8.3 | **7.3** | 8.2 | **8.6** |

Table 4: **Camera tracking results on *ScanNet*.** The metric is ATE and the unit is [cm].

| Iterations | tracking | mapping |
|---|---|---|
| NICE-SLAM [7] | 200 | 60 |
| Co-SLAM [8] | **20** | **30** |
| ESLAM [9] | 200 | 60 |
| Point-SLAM [31] | 200 | 150 |
| SplaTAM [16] | 200 | **30** |
| GS-SLAM [17] | 100 | 81 |
| DG-SLAM(Ours) | **20** | 40 |

Table 5: **Running iterations on *TUM*.**

## 4.4 Ablation study

To evaluate the efficacy of the proposed algorithm in our system, we conducted ablation studies across seven representative sequences from the *BONN* dataset. Given that the semantic priors within the *TUM* dataset have covered major motion categories (humans) while motion categories in *BONN* contained undefined dynamic objects such as balloons and boxes, thus we opted to perform ablation studies on the *BONN* dataset. We calculated the average ATE and STD metrics to illustrate the impact of various components on the overall system performance. As the results presented in Tab. 6, the findings affirm the effectiveness of all proposed methods in improving camera tracking capabilities. Specifically, the strategies for adding and pruning points are equally crucial, significantly impacting the quality of the Gaussian map reconstruction and, in turn, affecting the tracking performance. The depth warp operation effectively removes floaters from multiple viewpoints, thereby noticeably

enhancing the quality of the rendered images. One of the main contributions comes from the hybrid camera tracking strategy, which indirectly underscores the importance of eliminating inconsistencies between pose estimation and map reconstruction.

### 4.5 Time consumption analysis

As shown in Tab. 7, we report time consumption (per frame) of the tracking and mapping without computing semantic segmentation. These results were achieved through an identical experimental setup that involved conducting 20 iterations for tracking and 40 iterations for mapping, all processed on an RTX 3090Ti GPU. Benefiting from the rapid execution speed of Droid-VO and fast rendering of 3D Gaussian Splatting, our method exhibits a leading edge in terms of time consumption during the tracking process. Although our approach does not match the mapping speed of Co-SLAM, it achieves high-quality mapping and high-fidelity rendering with a competitive mapping running time. We also evaluate the inference time of our used semantic segmentation network, which required 163ms for every frame. It should be noted that our approach does not focus on the specific semantic segmentation network used, but rather on the fusion method itself.

|  | ATE[cm]↓ | STD[cm]↓ |
|---|---|---|
| w/o Add | 6.63 | 3.08 |
| w/o Prune | 6.89 | 3.16 |
| w/o Dep. Warp | 6.40 | 3.22 |
| w/o Seg | 15.27 | 7.47 |
| w/o Fine Tracking | 7.36 | 3.62 |
| DG-SLAM(Ours) | **5.51** | **2.79** |

Table 6: **Ablation study on *BONN* dataset.**

|  | Tracking [ms]↓ | Mapping [ms]↓ | Avg. Running [ms]↓ |
|---|---|---|---|
| NICE-SLAM [7] | 3186.2 | 1705.1 | 4892.3 |
| ESLAM [9] | 2045.9 | 1641.4 | 3688.5 |
| Point-SLAM [31] | 2279.5 | 1544.4 | 3823.9 |
| Co-SLAM [8] | 101.4 | **140.1** | **241.6** |
| DG-SLAM(Ours) | **89.2** | 549.3 | 645.9 |

Table 7: **Run-time comparison on *TUM* f3/w_s.**

## 5 Conclusion

In this paper, we have presented DG-SLAM, a robust dynamic Gaussian splatting SLAM with hybrid pose optimization under dynamic environments. Via motion mask filter strategy and coarse-to-fine camera tracking algorithm, our system significantly advances the accuracy and robustness of pose estimation within dynamic scenes. The proposed adaptive 3D Gaussians adding and pruning strategy effectively improves the quality of reconstructed maps and rendering images. We demonstrate its effectiveness in achieving state-of-the-art results in camera pose estimation, scene reconstruction, and novel-view synthesis in dynamic environments. While the tracking and reconstruction of large-scale scenes is currently the biggest limitation of our system, we believe it will be addressed by a more flexible loop-closure optimization algorithm in future work. Moreover, the accuracy of pose estimation of our system is still influenced by the segmentation precision of the semantic prior. Therefore, efficiently perceiving moving objects within the dynamic scenes remains an unresolved issue that warrants further exploration.

## Acknowledgement

This work is supported by National Key R&D Program of China (No.2019YFA0709502, No.2018YFC1312904), National Natural Science Foundation of China (Grant No.62106050 and 62376060), Natural Science Foundation of Shanghai (Grant No.22ZR1407500), ZJ Lab, Shanghai Center for Brain Science and Brain-Inspired Technology, and NIO University Programme (NIO UP).

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
