# OpenReview forum: "DG-SLAM: Robust Dynamic Gaussian Splatting SLAM with Hybrid Pose Optimization"
_NeurIPS.cc/2024/Conference — NeurIPS 2024 poster_

### Official Review · Reviewer_fGfv · 2024-06-13

**Soundness:** 3
**Presentation:** 3
**Contribution:** 3
**Rating:** 5
**Confidence:** 4

**Summary:**

This paper proposed a dynamic RGB-D SLAM system based on 3D Gaussian Splatting (3D-GS) and DROID-SLAM. Semantic segmentation masks and depth warping residuals are used to generate motion masks to remove the dynamic part of the scene. Experiments on three real-world datasets show the proposed approach achieves superior results on both dynamic and static scenes.

**Strengths:**

1. The paper is well written, technically sound and easy to follow.
2. The idea of using depth warping residuals as a complement to semantic priors for motion mask generation is simple but works effectively well.
3. The proposed method is very well engineered and achieved promising results on real-world dynamic sequences.
4. The proposed method is the first dynamic SLAM method based on 3D Gaussian Splatting (3DGS).

**Weaknesses:**

1. The technical novelty is a bit limited. The paper focuses on adding new components to previous SLAM methods in order to achieve dynamic SLAM.
2. The hybrid coarse-to-fine camera tracking seems like a trick: The coarse stage directly comes from DROID-SLAM while the fine stage simply adds masks to 3DGS-based tracking.

**Questions:**

Please see Weakness.

**Limitations:**

Yes.

---

> ### Author Rebuttal · Authors · 2024-08-07
>
> ### Q1: Technical novelty.
> A1: Thank you for your advice. We aim to clarify your concerns from the following perspectives: (1) As dynamic 3D Gaussian-based SLAM remains in its nascent stages of research and diverges markedly from traditional approaches, this paper presents preliminary explorations in this domain. We are the first work aimed at addressing the challenge of robust pose estimation for 3D Gaussian-based SLAM in dynamic environments. We hope to design an innovative dynamic SLAM system to better utilize 3D Gaussian explicit representation. (2) We have discovered that directly employing a 3D Gaussian representation for pose optimization in dynamic environments tends to result in convergence to local optima, and the solution process is notably unstable. Thus, we need to build a relevant robust tracking front end. (3) We design the motion mask generation method to filter out the invalid optical flow correspondence in Droid-VO to improve the robustness of the original tracking process. Furthermore, we have observed that Gaussian representations, when initialized with stable values, demonstrate improved convergence behavior. (4) Based on these observations and analyses, we propose a hybrid optimization approach incorporating a coarse-to-fine pose optimization strategy. Specifically, we employ a rectified confidence matrix within the droid-slam framework to provide initial pose estimates, which are subsequently refined using a global Gaussian map. This suite of designs significantly enhances the accuracy and robustness of pose estimation for GS-based SLAM in dynamic environments.
>
> ### Q2: The meaning and motivation of designing a hybrid coarse-to-fine strategy.
> A2: We sincerely appreciate the reviewer's meticulous and detailed review of our work, which helped us improve the quality of this paper. It should be noted that the coarse-to-fine approach is an optimization thought that has proven effective in enhancing the accuracy and robustness of pose estimation for complex tasks. However, how to achieve precise and robust pose optimization during the coarse and fine stages continues to be an open and challenging problem.
>
> We have introduced a hybrid pose optimization strategy specifically tailored for Gaussian-based SLAM systems to effectively tackle pose estimation challenges in dynamic environments, which also represents a central innovative contribution of our paper. Specifically, **in the coarse-optimized stage**, we have implemented a strategy that filters out unreliable and inaccurate optical flow estimations within the existing confidence matrix component of the original droid-slam system. This strategy removes inaccurate matching correspondences, thereby providing a relatively precise initial pose estimate. **During the fine-optimized stage**, in addition to utilizing motion mask generation methods to filter out invalid sampled points, we have designed a novel point management technique to maintain the Gaussian map. This encompasses adaptive point addition and effective outlier removal strategies. These elements are critical in ensuring that our SLAM system maintains high levels of accuracy and robustness in dynamic environments.
>
> Additionally, the coarse-to-fine pose optimization strategy **forms a compact and integral feedback mechanism within the entire SLAM system**. Improved pose optimization precision facilitates the generation of more accurate motion masks, which in turn enhance the accuracy of the initial pose estimates obtained during the coarse optimized stage and lead to better pose optimization results in the fine optimized stage.
>
> Furthermore, the keyframe selection strategy, which is predicated on optical flow motion during the coarse-optimized stage, further enhances the efficiency and accuracy of pose optimization in the fine-optimized stage. This integrated approach ensures a compact and effective optimization process throughout the whole SLAM system.

---

> > ### Comment · Area_Chair_K5ui · 2024-08-09
> >
> > Dear Reviewer  fGfv,
> >
> > Many thanks for your support. Could you please read this rebuttal and the below? Then give your responses? And any discussions are welcome.
> >
> > AC

---

### Official Review · Reviewer_MuLC · 2024-07-07

**Soundness:** 2
**Presentation:** 3
**Contribution:** 2
**Rating:** 5
**Confidence:** 4

**Summary:**

The paper introduces the first robust dynamic visual SLAM system based on 3D Gaussian Splatting. It provides precise camera pose estimation and high-fidelity reconstructions. Strategies such as motion mask generation, adaptive Gaussian point management, and hybrid camera tracking are proposed to improve the accuracy and robustness of pose estimation.

**Strengths:**

The writing is clear and easy to understand.
The paper proposed the first GS-based SLAM systems for dynamic scenes.
Compared with previous NeRF-based and GS-based SLAM systems, it achieves optimal performance on dynamic scenes.

**Weaknesses:**

1. The evaluation conducted in the paper primarily compares the proposed method against NeRF or Gaussian-splatting-based SLAM systems, which is not sufficient. It would be beneficial to include traditional SLAM systems as baselines, which present a more robust tracking capability, such as ReFusion [1] and ORB-SLAM3 [2].

2. Lacks the comparison with MonoGS [3] as the baseline of GS-based SLAM. MonoGS demonstrates more advanced map reconstruction ability compared with other GS-based alternatives, and presents robust camera tracking utilizing Jacobian optimization (e.g. compared with trivial pose optimization via photometric loss implemented in SplaTAM). It should be included as a baseline method for more exhausted comparisons.

3. Lacks survey on the literature of related work, such as Crowd-SLAM [4] and DDN-SLAM [5], which were also proposed for the same motivation of reconstruction on the dynamic environment.


4. The experiments on two datasets have shown the effectiveness (see point 5) of the proposed method in dynamic indoor settings. However, for a SLAM system, it would be preferable to conduct experiments on real-world scenes, especially in outdoor environments.

5. The proposed system largely relies on dynamic object removal in the tracking system; however, it lacks certain optimization for the mapping system. This results in the reconstructed scenes still containing artifacts.


[1] Palazzolo, J. Behley, P. Lottes, P. Giguere, and C. Stachniss, “Refusion: 3d reconstruction
in dynamic environments for rgb-d cameras exploiting residuals,” in IROS, 2019.

[2] C. Campos, R. Elvira, J. J. G. Rodr ́ıguez, J. M. Montiel, and J. D. Tard ́os, “Orb-slam3: An
accurate open-source library for visual, visual–inertial, and multimap slam,” IEEE TRO, 2021.

[3] Matsuki, Hidenobu, et al. "Gaussian splatting slam." Proceedings of the IEEE/CVF Conference on Computer Vision and Pattern Recognition. 2024.

[4] Soares, João Carlos Virgolino, Marcelo Gattass, and Marco Antonio Meggiolaro. "Crowd-SLAM: visual SLAM towards crowded environments using object detection." Journal of Intelligent & Robotic Systems 102.2 (2021): 50.

[5] Li, Mingrui, et al. "Ddn-slam: Real-time dense dynamic neural implicit slam with joint semantic encoding." arXiv preprint arXiv:2401.01545 (2024).

**Questions:**

1. According to Table 6, the segmentation mask seems to play a much more significant role than other components, such as the depth warp mask. Please discuss the utility of the other components in detail. A concern here is that a more accurate segmentation mask (e.g. produced by SAM) might facilitate even better dynamic removal without the need for other components.

2. In the implementation detail, the author should elaborate more on how to perform semantic segmentation, the parameter of depth warping, and the keyframe selection strategy they have integrated in the proposed system.

3. The dynamic features are removed in the tracking system; however, a large portion of the removed features might reduce the tracking accuracy. In other words, to what extent can the method proposed in the paper tolerate motion? Is there any failure case/example of tracking, e.g. when dynamic features dominate the current view?

4. The semantic mask is retrieved by utilizing a pretrained segmentation model. There is a concern regarding the generalizability of this semantic mask. How is the dynamic object defined with respect to the semantic mask? For instance, the human might be considered a dynamic object in the BONN dataset, but might be stable in other scenes (e.g. wax figure). In addition, how does this integration of the semantic model affect the real-time processing capability of the SLAM system?

**Limitations:**

There is no societal impact of this work, and limitations are discussed in the paper.

---

> ### Author Rebuttal · Authors · 2024-08-07
>
> ### W1: More experements results.
>
> A1: Thank you for your suggestion. We have added experiments comparing ORB-SLAM3[2] and Refusion[1]. As shown in Table 4 (one-page global PDF), our method presents a more robust tracking capability.
>
> ### W2: Comparing with MonoGS.
>
> A2: Thank you for your suggestion. Actually, we have already compared our method with MonoGS[3] in the initial draft. MonoGS is the name of the open-source framework and we used the name of its paper, namely GS-SLAM, as shown in Table2,3,4.  We will make modifications to this issue to prevent misunderstandings.
>
> ### W3: Related work.
>
> A3: Thank you for your suggestion. We will discuss and cite Crowd-SLAM[4] and DDN-SLAM[5] in the revised version.
>
> ### W4: Real-world scenes experiments.
> A4: Extending dynamic 3DGS to large-scale outdoor scenarios has always been a challenging problem. As mentioned in the conclusion part, the problem of loop closure for handling large-scale scenes is an interesting direction for future research. Therefore, we did not provide experimental results for outdoor environments.
>
> However, we provided experimental results for real-world scenes, as shown in Fig. 2 and Tab. 3(one-page global PDF). We conduct experiments on 3 real-world sequences and all results indicate the superiority of our method.
>
> ### W5: Lack certain optimization for the mapping system.
> A5: We have designed an adaptive Gaussian point addition and pruning strategy for the mapping system. We will perform geometric consistency verification on map points through depth warp. Consequently, artifacts can be eliminated to a certain extent. Of course, the constructed Gaussian map may still contain a small number of artifacts, and we are considering how to better resolve this issue.
>
> ### Q1: The role of the semantic mask. According to Table 6, the segmentation mask is much more important than other components.
> A1: In response to your observation that the segmentation mask seems to play a much more significant role than other components, we believe it relates to the dynamic object classes in the validation datasets, such as TUM and BONN. The semantic prior predominantly features dynamic object classes, such as humans, enhancing semantic segmentation performance. For non-rigid objects like humans, the results of depth warp fusion are less accurate compared to semantic segmentation, which is reasonable.
>
> Following your suggestion, we consider that SAM methods can indeed generate more accurate segmentation masks, provided that they receive inputs that are precise and unambiguous, such as points, boxes, or masks.
> While SAM methods could serve as an alternative approach to semantic segmentation, they still do not capture the true motion of objects. Therefore, a geometric consistency module is necessary for additional verification. Moreover, the inference speed of SAM methods is slower compared to current semantic segmentation approaches. Your suggestion has opened up new avenues for our research, and we are considering how best to integrate the SAM method into our SLAM system to improve the accuracy of our motion segmentation.
>
> ### Q2: More implementation detail.
> A2: Thank you for your helpful advice. We will enhance the description of the experimental details in the implementation section. We utilize OneFormer to generate prior semantic segmentation. For the depth wrap mask, we set the window size to 4 and the depth threshold to 0.6. For keyframe selection, we adopt the keyframe selection strategy from DROID-VO based on optical flow.
>
> ### Q3: To what extent can the method proposed in the paper tolerate motion?
> A3: As you mentioned, removing a large number of features could compromise tracking accuracy. Therefore, instead of employing a feature-based SLAM method as the front end, we utilize dense optical flow tracking from droid-SLAM for motion estimation. This choice enables our method to better tolerate dynamic environments. It can operate robustly and accurately in both low and high dynamic sequences of public datasets (TUM and BONN), as well as in real-world settings (Self-collected dataset). However, if occlusions are extensive, covering more than two-thirds of the entire image, the system is still prone to track loss. This issue remains a significant challenge for current SLAM systems and is yet to be resolved.
>
> ### Q4: The definition of the dynamic object with respect to the semantic mask and the running time of the semantic model.
> A4: Thank you for your suggestions. It is important to note that semantic segmentation is just one of the methods we use to generate motion masks; we also employ geometric consistency checks with depth warping operation. For different real-world applications, we typically predefine certain dynamic categories and initially obtain motion masks through semantic segmentation. In the vast majority of daily scenarios, humans are generally considered dynamic objects. Of course, the special scenario you mentioned, involving wax figures, may indeed struggle to balance predetermined motion priors with actual movements. However, such scenarios are relatively rare compared to typical applications, and we have not yet considered these cases in our approach. Regarding the real-time aspect of semantic segmentation you mentioned, it is essential to emphasize that our method does not focus on the specific architecture of the semantic segmentation network used, but rather on the design of information fusion methods. With ongoing advancements in these research fields and improvements in computing power, the processing speeds for semantic segmentation are expected to increase, ensuring they do not become bottlenecks for our method.

---

> > ### Comment · Reviewer_MuLC · 2024-08-09
> >
> > I appreciate the response from the authors, and my concerns are either reasonably addressed or considered as future works. Hence I raise my score to Borderline Accept.

---

> > > ### Author Response · Authors · 2024-08-12
> > >
> > > Dear Reviewer MuLC
> > >
> > > We appreciate the reviewer's time for reviewing and thanks again for the valuable comments.
> > >
> > > Best wishes
> > >
> > > Authors

---

### Official Review · Reviewer_DPxX · 2024-07-10

**Soundness:** 2
**Presentation:** 2
**Contribution:** 2
**Rating:** 6
**Confidence:** 3

**Summary:**

The paper present DG_SLAM to address the inconsistent observation of geometry and photometry in dynamic environments among the current slam-related field.  Specifically, the authors develop several technologies such as the motion mask generation, adaptive gaussian point management, and a hybrid camera tracking algorithm to improve the performance of slam system. And the experiments on TUM RGB_D and BONN RGB_D dataset showcases the state-of-the-art performance both on static and dynamic environments.

**Strengths:**

1.	For the task of motion object segmentation, the authors propose an advanced motion task generation method which integrates spatio-temporal consistent depth masks combined with semantic priors, and several rigorous mathematic formulations are illustrated to demonstrate the effectiveness.
2.	A coarse-to-fine camera pose optimization is proposed to optimize the whole slam system, and improve the constancy between pose and reconstructed 3d map.
3.	The authors also present an adaptive gaussian point addition and pruning strategy to manage the generated gaussian map.

**Weaknesses:**

1.	A motion mask generation strategy is used to mitigate the potential impact of observation noise from a single warp mask, can the authors give more details about this part? Are there any theoretically analysis for that?

2.	A hand-crafted rule is adapted in the map point deleting stage. For example, the authors delete points based on three baselines: the opacity value, the maximum scale, the ration between ellipsoid’s major and minor axes, can authors give more explanation about these criteria? Moreover, the authors showcase the removement of number of observations of the point that are too low, but fail to show its quantitative criteria.
3.	the running time on BONN dataset during the mapping phase seems a bit large in table 7, can authors give several details about this result? Are there any connections with the proposed map point deleting strategy?

**Questions:**

see weakness part

**Limitations:**

The method fails to be conducted on several large-scale real scenes.

---

> ### Author Rebuttal · Authors · 2024-08-07
>
> ### Q1: The details of the motion mask generation strategy.
> A1: During our experiments, we observed that inaccuracies in pose estimation, coupled with noise in the captured ground-truth depth values, can result in unstable motion segmentation outcomes when relying solely on a single warp mask. However, by integrating observations from depth warps across multiple frames, a more accurate motion mask can be constructed. This approach significantly enhances the reliability and precision of the motion segmentation process, as shown in Fig. 1 (one-page global PDF).
>
> ### Q2: The design criteria of the three gaussian point deletion methods.
> A2: Thank you for your suggestions. We aim to clarify your concerns from the following perspectives:
>
> (1) **The opacity value** denotes the importance of each gaussian point along each ray during the rendering process. Our opacity verification operation is similar in spirit to the original 3D GS. Since we implement an adaptive dense control algorithm to regulate the density of the Gaussian map, we contend that the deletion of non-essential Gaussian points can facilitate the incorporation of new Gaussian points and enhance the convergence of pose optimization.
>
> (2) Through the visualization of reconstructed Gaussian maps, we have observed that the Gaussian spheres with **large scale do not effectively represent the local details of objects**; instead, they tend to cause blurring or shadowing during the rendering process. This results in highly unstable pose optimization, which in turn leads to the failure of camera tracking.
>
> (3) The third criterion for evaluation arises because we have found that the rasterization of 3DGS imposes **no constraints on the Gaussians along the viewing ray direction**, even when a depth observation is present. This does not pose a problem when a sufficient number of carefully selected viewpoints are available. However, in continuous SLAM, this lack of constraint leads to numerous artifacts, which complicates tracking. Consequently, it becomes necessary to remove these Gaussian points.
>
> For each added Gaussian point, only a sufficient number of observations can confirm that the point is not an artifact. Based on this design principle, we select half the window size as the empirical threshold for point deletion operations. If a point is observed too infrequently, we consider it not robust enough and, therefore, it needs to be removed promptly.
>
> ### Q3: The details of the mapping processing running time on BONN dataset.
> A3: Your understanding is correct. The extended execution time of the mapping process on the Bonn dataset is closely linked to the point management strategy. More specifically, it is intricately associated with the extent of camera movement within the dataset.
>
> Due to the relatively large movements of the camera in the Bonn dataset, an increased number of Gaussian points are inserted during the mapping process to maintain tracking stability. Consequently, this leads to a noticeable augmentation in processing time during the mapping phase, as shown in Tab.2 (one-page global PDF). Even when addressing complex pose estimation issues in dynamic scenes, the runtime of our mapping process is approximately the same as that of other GS-based SLAM methods.

---

> > ### Comment · Reviewer_DPxX · 2024-08-08
> >
> > Thanks, the authors have processed all my concerns. I keep my rate unchanged.

---

> > > ### Author Response · Authors · 2024-08-12
> > >
> > > Dear Reviewer DPxX
> > >
> > > We appreciate the reviewer's time for reviewing and thanks again for the valuable comments.
> > >
> > > Best wishes
> > >
> > > Authors

---

### Official Review · Reviewer_UDnM · 2024-07-17

**Soundness:** 2
**Presentation:** 1
**Contribution:** 2
**Rating:** 3
**Confidence:** 5

**Summary:**

The paper combines existing methods for deep SLAM (DROID-SLAM) and 3D Gaussian splatting for 3D mapping into a system and adds an approach for dynamics filtering by depth warping to make tracking more robust in dynamic scenes. The approach is evaluated on several dynamic SLAM benchmarks and compared with recent deep learning based RGB-D SLAM approaches.

**Strengths:**

- The proposed approach of dynamics filtering for SLAM with 3D Gaussian splatting for map representation seems novel.
- Experiments demonstrate improvements over previous methods for neural RGB-D SLAM which have been designed for static scenes.

**Weaknesses:**

- Unfortunately, this paper has significant shortcomings in writing quality and clarity. Many variables and concepts are not defined properly (for instance, J, R in eq 2., \otimes operator, I_{m\times n}, \Pi in eq 8, S_i in l. 192, the symbols in eq. 13).
- Some phrases in the introduction are not accurate or proper English. What does "finishing pose estimation robustness" mean? What are invalid zones, what does furnishing an initial pose estimate mean?
- Sec. 2, the first paragraph starts with the heading "traditional visual SLAM with dynamic objects filter", but contains recent deep SLAM approaches for static scenes. Please restructure.
- The raycasted color and depth in eq. 3 requires a specific ordering of the gaussian splats, as indicated in the products ranging from j=1 to i-1. Please explain how it is established.
- What does the intersection operation over pixels achieve in eq 6 and 9? What does it mean to apply an indicator function and take a \otimes product with the I_mxn ? What is I_mxn?
- The variable \mu is used for coordinates and means of 3D Gaussians ambiguously, please change notations.
- l. 179, how is the camera pose parametrized and optimized ?
- l. 188, what is the dynamic radius?
- l. 192, what is the scale vector S_i and why should it be computed from the mean distance of three nearest neighbor points? Where do these points come from?
- eq 12, how are the lambda_1-3 chosen?
- l. 235, "unit quaternion algorithm" is not the right name for the method but rather Horn's procrustes method.
- l. 257, what is an "analytical experiment" ?
- Table 2, 3, what are the metrics for the numbers ? ATE, RPE ? how is it computed?
- Tables 2, and 3 should also compare with classical dynamic SLAM methods like Co-Fusion [*1], MID-Fusion [*2], EM-Fusion [*3] to name a few. Also these methods are missing from the discussion of related work.
- what is an "invalid Gaussian point filter" in l. 272 ?
- Table 5 seems to indicate that different settings of the iteration count hyperparameter are needed for the various datasets. This indicates an issue with generalization for the method. Are other hyperparameters chosen consistently across datasets?
- l. 290 what does to "encompass major motion categories" mean ?
- l. 291, what is the definition of the STD metric ?
- Table 7 / l. 302, please include run-time for semantic segmentation as it seems essential according to Table 6.
- How is the semantic segmentation mask obtained ? What is the runtime of this processing step?
- The paper does not discuss limitations and assumptions of the method beyond not performing loop closing.

[*1] Martin Ruenz, Lourdes Agapito. Co-fusion: Real-time segmentation, tracking and fusion of multiple objects. ICRA 2017

[*2] Binbin Xu, Wenbin Li, Dimos Tzoumanikas, Michael Bloesch, Andrew J. Davison, Stefan Leutenegger. MID-Fusion: Octree-based Object-Level Multi-Instance Dynamic SLAM. ICRA 2019

[*3] Michael Strecke, Joerg Stueckler. EM-Fusion: Dynamic Object-Level SLAM With Probabilistic Data Association. ICCV 2019

**Questions:**

See questions in paper weaknesses.

**Limitations:**

The conclusion provides an obvious limitation of the approach due to not considering loop closing. Further limitations and assumptions are not discussed.

---

> ### Author Rebuttal · Authors · 2024-08-07
>
> We sincerely appreciate the reviewer's meticulous and detailed review of our work, which helped us improve the quality of this paper. It should be noted that some of the issues you raised are typically treated as foundational definitions in the 3D Gaussian Splatting paper and traditional SLAM literature. To avoid unnecessary repetition, we have omitted some of the common expressions and explanations of symbols in our previous papers.
>
> A1: (1)Eq.2 comes from the 3DGS[1] paper. $R$ is the viewing transformation and $J$ is the Jacobian of the affine approximation of the projective transformation. (2) $I_{m\times n}$ represents a matrix of the same size as the image, filled with ones. (3) Eq.8 is derived from DROID[2], where $p_i$ means a grid of pixel coordinates. (4) $S_i$ in l.192 and $R_i$ in l.193 are scaling vector and rotation matrix respectively. (5) In Eq.13, $\alpha_{i}$ means the opacity, which has been defined in l.125, and $S$ means the scaling vector. Other threshold parameters can be find more details in l.244.
>
> A2: (1)"finishing pose estimation robustness" means achieving robust pose estimation. (2)“invalid zones” means dynamic object regions, illegal depth value regions, and regions with less rendering opacity. (3)"furnishing an initial pose estimate" represents that the coarse stage provides a better initial camera pose for the fine stage.
>
> A3: We will modify it to "visual SLAM with dynamic objects filter".
>
> A4: This has been explained in 3DGS[1]. The raycasting will sort all the Gaussian points based on their depth value.
>
> A5: For the defination of $I_{m\times n}$, please refer to A1. The intersection operation signifies that for each element in the matrix $I_{m\times n}$, we assess whether its warp depth meets a specified threshold and subsequently modify the corresponding value at that position.
>
> A6: The mean of  3D Gaussian points in space means the center coordinates of the Gaussian points. Thus, we use $\mu$ to represent the coordinates and mean. Their meanings are consistent.
>
> A7: We parametrize the camera pose using quaternion and translation. To efficiently execute the pose optimization process, we have computed the Jacobian of the designed loss function to these two optimization variables and have implemented modifications in CUDA.
>
> A8: We employed the dynamic point density management strategy, which determines whether to insert new Gaussian points according to the dynamic radius. The dynamic radius is determined based on color gradient, as explained in l.202 - l.204.
>
> A9: $S_i$ refers to the scale of each Gaussian point, initialized by the mean distance of three nearest neighbor points. This operation originated from the original 3DGS[1] paper.
>
> A10: $\lambda_1$-$\lambda_3$ are the loss weight, determined through grid search. We show the details of these parameters in l.242.
>
> A11: Thank you for your advice. We will update this expression in the revised paper.
>
> A12: We utilized the GT point cloud provided by the Bonn to qualitatively and quantitatively evaluate the reconstruction results, where "quantitative analytical experiments" refer to the results in Tab.1.
>
> A13: The values presented in Tab. 2 and 3 correspond to the Absolute Trajectory Error (ATE) metrics, as mentioned in the Metrics section of the experiments (l.233). ATE quantifies the average absolute trajectory error between the estimated and ground truth poses across all measurement locations. Further details can be found in the Evo evaluation tool.
>
> A14: Thank you for your suggestion. In the experimental section, we have added and compared some classic dynamic SLAM methods: Co-Fusion[4], MID-Fusion[5], and EM-Fusion[6].  The results are shown in Tab. 1(one-page PDF). Our method has showcased superior pose estimation results in comparison to classic dynamic SLAM methods. We will also discuss these methods in related work.
>
> A15: It means “Map point deleting strategy” mentioned in l.214, Sec 3.4.
>
> A16: In Table 5, we present the number of iterations of the tracking and mapping process for different SLAM methods. The experimental results show that our method can achieve superior results with fewer iterations. What's more, together with Table 7, Table 5 also demonstrates the competitiveness of our method in time consumption.
>
> A17: We divided some default dynamic object classes based on semantic prior, such as humans. However, the semantic mask requires prior knowledge of motion categories and faces generalization challenges. Thus we use the depth warp mask to identify undefined dynamic objects such as balloons and boxes.
>
> "encompass major motion categories" means the dynamic object category contained in Tum is human. Therefore, we opted to perform ablation studies on BONN. It contains moving balloons and boxes, which can show the effectiveness of our depth warp mask.
>
> A18: STD means Standard Deviation of Absolute Trajectory Error. This concept has been mentioned in l.234. Further details can be found in the Evo evaluation tool.
>
> A19: The semantic segmentation process can be considered as a part of data preprocessing. Thus, it has not been included in the system runtime calculations. To address your concern, we have tested the inference time of the Oneformer[7] method we adopted on an A6000 GPU, which is 163ms for every frame. It should be noted that our approach does not focus on the specific semantic segmentation network used, but rather on the fusion method itself.
>
> A20: Thank you for your helpful advice. We will enhance the description of the experimental details in the implementation section. We utilize OneFormer[7] to generate prior semantic segmentation. We reported the run-time in A19.
>
> A21: Extending dynamic 3DGS to large-scale scenarios has always been a challenging problem. As mentioned in GS-SLAM[8], the problem of loop closure for handling large-scale scenes is an interesting direction for future research. Thus, we believe it is still a challenge worth solving in dynamic SLAM.

---

> > ### Comment · Reviewer_UDnM · 2024-08-10
> >
> > Thanks for the detailed response to my comments.
> >
> > Some further comments/questions:
> > * Even if the methods have been introduced in previous papers, used notation needs to be defined properly and in a self-contained way in a scientific paper to avoid any ambiguities. The paper should be revised accordingly.
> > * A7, which rotation representation is used for optimization and how are the constraints on SO(3) handled?
> > * A16, Table 7 results are from a different dataset than Table 5. Run-time results on the TUM datasets would be needed to relate the iterations with run-time. Please also include comparison with the classical methods [4-6]
> > * A19, the semantic segmentation is essential for the system and therefore the runtime cannot be omitted. Please include the timing in the run-time evaluation. Do the baseline methods in Table 7 also contain semantic segmentation? Is it counted into the run-time?
> > * A21, please also discuss limitations and assumption wrt. the design choices of the proposed method. For instance, how does the approach depend on the accuracy of semantic segmentation? Discuss if live processing of sensory streams is possible with this approach.

---

> ### Author Response · Authors · 2024-08-12
> **Response to Reviewer UDnM [1/2]**
>
> ### Q1: Notation issue.
> A1: We are grateful for the reviewer's valuable comments. We will revise and refine the notations in the revision to eliminate any potential ambiguities.
>
> ### Q2: Rotation representation and constraints within the SO(3) group.
> A2: We utilize the quaternion as the rotation representation in the optimizer to facilitate pose optimization. Throughout the optimization process for the rotational variables, we compute the derivative of the final loss directly with respect to the current quaternion representation. This chain rule differentiation process is structured into two stages. In the first stage, the derivative from the quaternion to the rotation matrix is calculated using automatic differentiation within Pytorch, which allows for the automatic capture of optimization variables. In the second stage, the derivative of the final loss with respect to the rotation matrix is manually computed using the Jacobian matrix. This calculation is implemented using CUDA code during the Gaussian rendering process. Upon obtaining the optimized quaternion, we normalize it to achieve a unit quaternion. This entire optimization process ensures compliance with the constraints on the SO(3) group.
>
> ### Q3: More experiments and consistent comparison.
> A3: Thank you for your suggestions. To address the reviewer's concern, we provide the running time of tracking and mapping on the TUM dataset, as shown in Tab. 1. All the experiments were conducted with 20 iterations for tracking and 40 iterations for mapping. We have also added a comparison with the classical methods: Co-Fusion and MID-Fusion, where the running time is cited from the original papers.
>
> **Table1: Run-time comparison on TUM f3_walk_static. * denotes including semantic segmentation time.**
> ||Tracking[ms]| Mapping[ms]|Avg[ms]|
> |:----:|:----:|:----:|:----:|
> |Co-Fusion|-|-|**83.3**|
> |MID-Fusion *|-|-|400.0|
> |NICE-SLAM|3186.2|1705.1|4892.3|
> |E-SLAM|2045.9|1641.4|3688.5|
> |Point-SLAM|2279.5|1544.4|3823.9|
> |Co-SLAM|101.4|**140.1**|241.6|
> |Ours|**89.2**|549.3|645.9|
>
> Classical methods utilize an efficient variant of the Iterative Closest Points (ICP) algorithm for camera tracking and avoid a training process for mapping. Moreover, the implementation of classical methods has been optimized through the use of a C++ framework, resulting in faster runtime compared to neural SLAM methods. Currently, both NeRF and GS-based SLAM methods exhibit a certain gap in runtime when compared to traditional approaches. Addressing this discrepancy is a challenge that the research community continues to actively pursue.

---

> ### Author Response · Authors · 2024-08-12
> **Response to Reviewer UDnM [2/2]**
>
> ### Q4: Running time issue.
> A4: It should be noted that our approach does not focus on the specific semantic segmentation network used, but rather on the fusion method itself. All the baseline methods in Table 7 in our main paper do not include the time for semantic segmentation. Semantic segmentation is typically employed as a preprocessing operation and serves as an input to the system. Most of the dynamic SLAM methods also did not include the time for semantic segmentation instead including it as part of the preprocessing step. Consequently, Table 7 captures the total runtime of our SLAM system, and the time taken for semantic segmentation is not included. Considering the reviewer's concerns about the runtime of the semantic segmentation network we utilized, we have tested the inference time of the Oneformer [7] method, which is 163ms for every frame. Correspondingly, the running time including semantic segmentation is 245.57ms for tracking.
>
> To avoid potential ambiguity, we will provide a detailed description of the running time per frame for the semantic segmentation method employed in the revised paper.
>
> Note that DG-SLAM is not optimized for real-time operation. As the first system to propose ***dynamic*** Gaussian splatting for SLAM, this paper primarily focuses on designing an effective approach to achieve robust tracking and mapping. With ongoing advancements in these research fields and improvements in computing power, the processing speeds for semantic segmentation are expected to increase, ensuring they do not become bottlenecks for our method.
>
> ### Q5: Discuss more limitations and assumptions.
> A5: Our approach has a certain tolerance for the accuracy of semantic segmentation. When the semantic segmentation model generates incomplete or incorrect segmentation masks for fewer frames, our system can still perform accurate and robust camera tracking. This is attributed to the coarse-to-fine pose optimization strategy where the motion estimation from the dense optical flow can tolerate some segmentation failures. Thus the coarse stage can provide a robust initial pose for the fine stage.
>
> Even when addressing complex pose estimation issues in dynamic scenes, the runtime of our mapping process is approximately the same as that of other GS-based SLAM methods, as shown in Table 2 (one-page global PDF). The system runtime reported in the original paper was evaluated on a 3090ti GPU. With ongoing advancements in GPU computational capabilities, we believe our method is capable of real-time computation, enabling the live processing of sensory data streams.
>
> Hope our response helps the reviewer's final recommendation. Thank you!

---

> ### Author Response · Authors · 2024-08-13
>
> Dear Reviewer UDnM,
>
> We appreciate your time for reviewing, and thanks again for the valuable comments and suggestions. As the discussion phase is nearing its end, we wondered if the reviewer might still have any concerns that we could address. We believe our responses on consistent comparison, rotation representation, running time, and limitations addressed all the questions/concerns, and we hope that our work’s impact and results are better highlighted with our responses.
>
> It would be great if the reviewer can kindly check our responses and provide feedback with further questions/concerns (if any). We would be more than happy to address them. Thank you！
>
> Best wishes,
>
> Authors

---

### Author Rebuttal · Authors · 2024-08-07

We are immensely grateful for the time and effort expended by all reviewers in reviewing our manuscript. The technical evaluations and detailed comments provided have been invaluable and have substantially enhanced the quality of our work. In this response, we have meticulously addressed each question posed by the reviewers on a point-by-point basis. Additionally, we have included the figures and tables from the supplementary experiments in the one-page PDF attachment.

The citation numbers in response to the reviewer UDnM are as follows:

[1] Bernhard Kerbl, Georgios Kopanas, Thomas Leimkühler, and George Drettakis. 3d gaussian splatting for real-time radiance field rendering. ACM TOG, 2023.

[2] Zachary Teed and Jia Deng. DROID-SLAM: Deep Visual SLAM for Monocular, Stereo, and
362 RGB-D Cameras. In NeurIPS, 2021.

[3] Erik Sandström, Yue Li, Luc Van Gool, and Martin R. Oswald. Point-slam: Dense neural point cloud-based slam. In ICCV, 2023.

[4] Martin Ruenz, Lourdes Agapito. Co-fusion: Real-time segmentation, tracking and fusion of multiple objects. In ICRA, 2017.

[5] Binbin Xu, Wenbin Li, Dimos Tzoumanikas, Michael Bloesch, Andrew J. Davison, Stefan Leutenegger. MID-Fusion: Octree-based Object-Level Multi-Instance Dynamic SLAM. In ICRA, 2019.

[6] Michael Strecke, Joerg Stueckler. EM-Fusion: Dynamic Object-Level SLAM With Probabilistic Data Association. In ICCV, 2019.

[7] J. Jain, J. Li, M. Chiu, A. Hassani, N. Orlov, and H. Shi, “OneFormer: One Transformer to Rule Universal Image Segmentation,” in CVPR,
2023.

[8] Hidenobu Matsuki, Riku Murai, Paul H. J. Kelly, and Andrew J. Davison. Gaussian Splatting SLAM. In CVPR, 2024.

---

### Decision · Program_Chairs · 2024-09-25

**Decision:**

Accept (poster)

**Comment:**

This paper proposes a DG-SLAM system in dynamic environments grounded in 3D Gaussians with fidelity reconstructions. The key parts include motion mask generation, adaptive Gaussian point management, and a hybrid camera tracking algorithm to improve the accuracy and robustness of pose estimation. Experimental results show the effectiveness of the method.

The contribution of the paper is to tackle dynamic objects when appearing in the environment. The existing neural SLAM methods do not perform well in dynamic scenes. The robustness of these systems significantly decreases, even leading to tracking failures. This paper proposes a 3D Gaussian-based visual SLAM that can reliably track camera motion in dynamic indoor environments.

The paper received one “Reject” rating, one “Weak Accept” rating, and two “Borderline Accept” ratings.

Reviewer UDnM giving “Reject” proposed a series of questions. The questions were addressed or discussed by the authors. In particular, the experiments were performed with classical methods Co-Fusion, MID-Fusion, EM-Fusion on accuracy in Tables 2 and 3, and performed with Co-Fusion, MID-Fusion on runtime. The authors discussed both NeRF and GS-based SLAM methods exhibit a certain gap in runtime when compared to traditional approaches. Addressing this discrepancy is a challenge that the research community continues to actively pursue.

Reviewer DPxX thinks the authors have processed all his/her concerns. Reviewer DPxX keeps the “Weak Accept”.

Reviewer MuLC thinks his/her concerns are either reasonably addressed or considered as future works. Reviewer MuLC raised the score to Borderline Accept.

Reviewer fGfv didn’t give replies to the authors’ rebuttals. Reviewer fGfv’s score is “Borderline Accept”.

In summary, the paper has new contributions and the questions of reviewers are addressed in the authors’ replies.

An expected revision is to see the experiments compared with ORBSLAM3 on accuracy. Comparisons on runtime are not required. ORBSLAM3 is also a dense RGBD SLAM and can deal with dynamic objects. Generally, many SLAM systems can deal with dynamic objects by removing the matchings on the dynamic objects. When the occupancies of the dynamic objects on images are larger, the camera pose tracking will fail. How about the proposed method at the time?

Another expected revision is to see the discussions on why the method can’t be used in outdoor environments. Although RGBD cameras fail to capture depths, Lidar-cameras still can be used.